# Knowledge, perception, and attitude toward premarital screening among university students in Kurdistan region–Iraq

**Kochr A. Mahmood**[1]*, **Govand S. Sadraldeen**[1], **Samir M. Othman**[2], **Nazar P. Shabila**[2,3], **Abubakir M. Saleh**[2,4], **Kameran H. Ismail**[2]

**1** Faculty of Medicine, Koya University, Koya, Kurdistan Region, Iraq, **2** Collage of Medicine, Hawler Medical University, Erbil, Kurdistan Region, Iraq, **3** Collage of Health sciences, Catholic University in Erbil, Erbil, Kurdistan Region, Iraq, **4** Collage of Nursing, Tishk International University, Erbil, Kurdistan Region, Iraq

\* Kochr.mahmood@koyauniversity.org

**Data Availability Statement:** All relevant data are within the paper and its Supporting Information files.

## Abstract

Premarital screening programs are essential for identifying and providing counseling to couples at risk of transmitting genetic diseases or sexually transmitted infections. Despite their importance, university students' awareness and knowledge of premarital screening programs remain inadequate. This study aimed to evaluate the knowledge, perceptions, and attitudes of university students in the Kurdistan Region of Iraq regarding premarital screening programs. A cross-sectional survey involving 960 students was conducted from December 2023 to February 2024. The survey assessed participants' demographics, knowledge, perception, and attitudes toward PMSP using a structured questionnaire. Findings revealed that a significant portion of participants (39.4%) had poor knowledge of premarital screening programs, 35.9% had fair knowledge, and only 24.7% had good knowledge. Despite limited knowledge, there was strong support for premarital screening programs, with 83.1% agreeing on its importance and 78.8% recognizing the need for premarital awareness. Most participants (65.8%) believed premarital screening programs could reduce genetic diseases, and 65.6% thought it could lower sexually transmitted diseases' prevalence. Cultural acceptance of marrying relatives was notable, with 59.7% disagreeing with the preference for not marrying relatives. Married participants showed significantly higher knowledge and attitude scores compared to single participants. Gender differences were observed, with males having higher knowledge scores. There were no significant differences in perception and attitude scores based on gender or residential area. The study underscores the need for enhanced educational campaigns to improve premarital screening programs awareness and positively influence attitudes, especially targeting cultural aspects like accepting relative marriages. Comprehensive education and fostering positive attitudes toward premarital screening programs are vital for their broader acceptance and implementation.

**Funding:** The authors received no specific funding for this work.

**Competing interests:** The authors have declared that no competing interests exist.

## Introduction

Premarital screening (PMS) programs are critical public health tools designed to identify and provide counselling for couples at risk of transmitting genetic diseases to their offspring or who may be at risk of sexually transmitted infections. Despite their importance, university students' awareness and knowledge of these screenings remain a topic of interest and concern. This demographic represents a significant portion of society that is approaching or currently at the marriageable age, making their understanding of PMS vital for the well-being of future generations [1]. Consequently, many nations and health organizations have embarked on educational campaigns to shed light on the significance of premarital tests, covering everything from inheritable diseases to reproductive health. However, the effectiveness of these programs is contingent upon the degree to which university students are engaged and informed about the purpose, procedures, and benefits of PMS. Research into this area often seeks to evaluate the awareness levels of these students and to identify opportunities to enhance education and outreach initiatives to improve the uptake and positive outcomes of premarital health practices [2–4].

Tests for common genetic blood disorders (primary hemoglobinopathy, such as thalassemia and sickle cell anemia) and infectious diseases (like hepatitis B, hepatitis C, and HIV/AIDS) are referred to as PMS if a couple plans to get married soon [5]. PMS aims to give couples/partners options to help them plan a healthy family and medical advice regarding the likelihood of passing on the diseases above to the partner/spouse or other children. This process lowers the financial burden of treating sexually transmitted infections and some genetic disorders by reducing their prevalence. It lessens the workload for state-run hospitals and blood banks [6]. It also raises awareness of a virtuous and well-being marriage. In addition, premarital examinations may involve blood tests, resuscitation factors, semen analysis, FSH, prolactin, testosterone, and estrogen hormones, as well as tests for syphilis, gonorrhea, and other sexually transmitted diseases [7]. Understanding the factors influencing students' knowledge about PMS—including cultural, religious, and socioeconomic backgrounds- and the role of university curriculum and public health interventions—is crucial for developing targeted strategies to boost their awareness and encourage proactive health behaviors. These endeavors not only support public health but also foster a culture of prevention, which is instrumental in reducing the burden of genetic and communicable diseases on individuals, families, and healthcare systems worldwide [4,8,9].

Premarital counseling is a type of therapy designed to help couples become better prepared for marriage, recognize issues in their relationship, and give them the tools to handle disagreements both now and in the future. Couples also communicate their unique needs, preferences, and expectations for their marriage and learn how to settle disagreements in ways that satisfy both parties [10]. In three Kurdistan provinces in northern Iraq (Duhok, Erbil, and Sulaimaniyah), a PMS program to find hemoglobinopathy carriers was started in 2008. The passage of legislation by the regional parliament mandating PMS for hemoglobinopathies made the program easier to implement [11]. Possibly public people and university students may have information about premarital screening test, while there is no enough evidence to proof that. Because of their age of conception, level of education, and potential for early health interventions that could have long-term benefits for public health, university students were selected for a study on premarital screening tests.

The current study aimed to determine students' knowledge, perception, and attitude regarding PMS programs and find the important factors related to their knowledge.

## Methods

### Design and setting

This cross-sectional study was conducted in Kurdistan Region of Iraq, from (20/12/2023) to (25/2/2024). The survey was based on a self-administered online survey using Google Forms.

## Participants

The sample size for this study was calculated using Epi-info, using an estimated prevalence of household members who need care of 33.9%, with a confidence interval of 95% and ±5% precision. Thus, a sample size of 344 individuals was calculated, which was increased to 500 to account for nonresponse. However, 960 students answered the questions. A convenience sample of students from the universities was recruited for this study.

## Study tool and data collection

The questionnaire used for data collection included three sections. The questionnaire previously has been used [5,9].The first section included information on participants' characteristics, such as age, gender, and economic status, and questions about hereditary diseases inside their families. The second section included participants' knowledge of the PMS test, which included nine questions to assess their knowledge. All the questions were answered with yes or no. In the third section, all the questions were related to the perception of the participants for PMS, and all of them were answered with disagree, neutral, and agree. This section included 12 questions to evaluate their perception. The final section included questions related to participants' attitudes toward PMS. It contained four questions and answered with disagree, neutral, and agree. For all three sections of the questions, knowledge, perception, and attitude, less than 50% considered poor, 50%-75% considered fair, and more than 75% considered good [12].

The online Google form was shared with the participants through social media, WhatsApp groups, and email. The form began with a description of the study and its significance. Participants were asked to provide informed consent before completing the online Google form.

## Ethical statement

Participants provided written informed consent online after being informed that participation was voluntary and anonymity was guaranteed. The Research Ethics Committee of the authors' institute approved the study protocol. Koya University (2/3/17-09-2023) and Hawler Medical University (1/12/5-9-2023).

## Data processing and statistical analysis

Different statistical tests were used to determine the factors that affected three elements of the study. Kolmogorov Smirnov test was used to determine the normality of the continuous variables. Since the data was non-parametric, the Mann-Whitney U and Kruskal Walles tests were used, and mean rank and mean with standard deviation were used to present the data. The data were analyzed using the Statistical Package for the Social Sciences (version 27). A p-value equal to and less than 0.05 was considered to indicate statistical significance.

## Results

In general, 960 persons answered the questions. 394 (41.1%) and 408 (42.5%) were less than 20 and 20 to 22 years old, respectively. The majority of the participants were female (n = 582, 60.6%). Most of them were single (n = 890 92.7%). Most (n = 907, 94.5%) were from urban areas, and 753 (78.4%) had enough economic status for life. Over half answered yes to parental consanguinity (n = 571, 59.5%). Regarding personal and family history for hereditary diseases, most participants reported no diseases (n = 814, 84.8% and n = 620, 64.6)), respectively. 319 (33.2) were from stage 4, and 280 (29.2) were from stage 1. The median and interquartile range was 3 (3), as shown in Table 1.

**Table 1. Sociodemographic characteristics of the participants.**

| Variable | Frequency | Percentage |
|---|---|---|
| Age group | | |
| Less than 20 | 394 | 41.1 |
| 20–22 | 408 | 42.5 |
| More than 22 | 157 | 16.4 |
| Gender | | |
| Female | 582 | 60.6 |
| Male | 378 | 39.4 |
| Marital status | | |
| Single | 890 | 92.7 |
| Married | 70 | 7.3 |
| Resident place | | |
| Urban area | 907 | 94.5 |
| Rural area | 53 | 5.5 |
| Economic status | | |
| Not enough for daily life | 79 | 8.2 |
| Enough for daily life | 753 | 78.4 |
| Good | 128 | 13.3 |
| Parental consanguinity | | |
| No | 389 | 40.5 |
| Yes | 571 | 59.5 |
| Personal history of hereditary diseases | | |
| No | 814 | 84.8 |
| Yes | 146 | 15.2 |
| Family history of hereditary diseases | | |
| No | 620 | 64.6 |
| Yes | 340 | 35.4 |
| In which stage do you study | | |
| 1 | 280 | 29.2 |
| 2 | 146 | 15.2 |
| 3 | 137 | 14.3 |
| 4 | 319 | 33.2 |
| 5 | 13 | 1.4 |
| 6 | 65 | 6.8 |
| Median | 3 | |
| Inter quartile range | 3 | |

Regarding the knowledge of the students, the vast majority of respondents have heard of PMS (n = 905, 94.3%) and know its meaning (n = 850, 88.5%) and objectives (n = 895, 93.2%). Furthermore, A majority recognize that PMS focuses on infectious (n = 739, 77%) and hereditary diseases (n = 863, 89.9%). Less than half of the respondents are aware of different places to perform PMS (n = 405, 42.2%), more than half (n = 507, 52.8%) know the specific tests included in PMS, and a similar proportion knows about the physical examinations included in PMS (n = 496, 51.7%), as shown in Table 2.

In terms of the sources of information that the students used, 477 (49.68%) of the participants took information from their family, 109 (11.4%) from the internet, 73 (7.6%) from school, and 301 (31.4%) from all sources.

**Table 2. Knowledge of the participants regarding PMS.**

| Questions | Frequency | Percentage |
|---|---|---|
| Did you hear PMS previously | | |
| No | 55 | 5.7 |
| Yes | 905 | 94.3 |
| Do you know the meaning of PMS | | |
| No | 110 | 11.5 |
| Yes | 850 | 88.5 |
| Do you know the objectives of PMS | | |
| No | 65 | 6.5 |
| Yes | 895 | 93.2 |
| PMS focuses on infectious diseases | | |
| No | 221 | 23 |
| Yes | 739 | 77 |
| PMS focuses on hereditary diseases | | |
| No | 97 | 10.1 |
| Yes | 863 | 89.9 |
| Do you know different options of places to perform PMS | | |
| No | 555 | 57.8 |
| Yes | 405 | 42.2 |
| Do you know the tests included in PMS | | |
| No | 453 | 47.2 |
| Yes | 507 | 52.8 |
| Physical examinations included in PMS | | |
| No | 464 | 48.3 |
| Yes | 496 | 51.7 |

As shown in Fig 1, most participants fall into the "Poor Knowledge" category (n = 378, 39.4%), followed by those with "Fair Knowledge" (n = 345, 35.9%), and the fewest have "Good Knowledge" (n = 237, 24.7%). As shown in graph 1, most participants fall into the "Poor Perception" category (n = 394, 41%), followed by those with "Fair Perception" (n = 346, 36%), and the fewest have "Good Perception" (n = 220, 22.9%). This indicates that the overall perception of the topic among participants is more negative than positive, with a considerable number of participants having only a fair or poor perception. Most participants (n = 776, 80.8%) exhibit a Poor Attitude. This category has the highest frequency by a significant margin, suggesting that most participants are not positively engaged or have negative feelings about the subject in question. A smaller but notable portion of participants (n = 155. 16.1%) have a Fair Attitude. Only 29 (3%) participants have a Good Attitude, indicating a very small fraction of the total population has a positive outlook.

The attitude of the participants for PMS. A large majority (n = 798, 83.1%), agree that PMS is important for future couples, indicating strong support for its perceived value. Additionally, most respondents (n = 756, 78.8%), agree that awareness about PMS before marriage is important, suggesting a high level of perceived importance for premarital awareness. A majority (n = 632, 65.8%) believe that PMS can reduce the prevalence of some genetic diseases, though a notable portion is neutral (n = 216, 22.5%) or disagrees (n = 112, 11.7%). A significant majority (n = 666, 69.4%), agreed that consanguinity increases the risk of hereditary diseases, while a substantial number were neutral (n = 288, 30%). A majority (n = 630, 65.6%) believed PMS would reduce the prevalence of some sexually transmitted diseases, but a considerable portion is neutral (n = 258, 26.9%). Most respondents (n = 558, 58.1%) agree on the confidentiality of

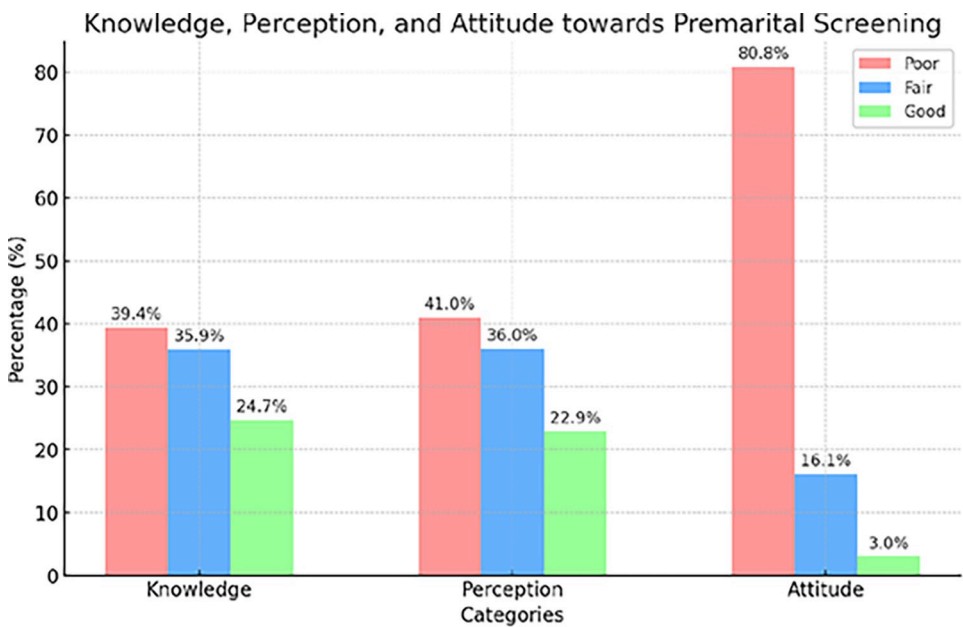

**Fig 1. Knowledge, perception and attitude of the students towards premarital screening.**

PMS and genetic counseling, though a significant number are neutral (n = 331, 34.5%). However, the majority are either neutral (n = 430, 44.8%) or disagree (n = 371, 38.6%) that PMS causes psychological troubles, indicating that most do not see it as a significant psychological burden. Moreover, the majority (n = 616, 64.2%) believe religious people should adopt PMS ideas in their discussions, showing support for integrating PMS into religious dialogue. Also, a significant majority (n = 755, 78.6%) support legal obligation for PMS and genetic counseling, indicating strong support for mandated screening. While opinions are more divided here, with 382 (39.8%) agreeing that the decision should be left to the couple, while a significant number are neutral (n = 345, 35.9%) or disagree (n = 223, 24.3%). A majority (n = 617, 64.3%) agree on the importance of medical counseling post-results, with a notable neutral stance (n = 276, 28.7%). Additionally, a significant majority (n = 674, 70.2%) agree that any disease detected should be treated before marriage, indicating strong support for premarital treatment. Table 4 shows the details of the perception of participants toward PMS (Table 3).

The percentages reflect the proportion of participants who disagree, are neutral or agree with each statement. A significant majority (n = 781, 81.4%) of participants agree with carrying out PMS, indicating strong support for this practice. Very few participants (n = 50, 5.2%) disagree, while a modest portion (n = 129, 13.4%) remain neutral. Most participants (n = 573, 59.7%) disagree with the preference for not marrying relatives, indicating a cultural or social acceptance of relative marriages. Only a small percentage (n = 69, 7.2%) agree with this preference, and a significant portion (n = 318, 33.1%) are neutral. Most participants (n = 788, 82.1%) agree that the appropriate time for PMS is just before marriage, showing strong consensus. A few (n = 40, 4.2%) disagree, and a small percentage (n = 132, 13.8%) are neutral.

A majority (n = 585, 60.9%) agree that future couples should be advised to conduct PMS, indicating general support for promoting PMS. However, a notable minority (n = 173, 18%) disagree, and (n = 202, 21%) are neutral, suggesting some reservations or lack of strong opinion among these groups (Table 4).

**Table 3. Perception of the participants toward PMS.**

| Question | Disagree (%) | Neutral (%) | Agree (%) |
|---|---|---|---|
| PMS is important for future couples | 57 (5.9) | 105 (10.9) | 798 (83.1) |
| Awareness about PMS before marriage is important | 53 (5.5) | 151 (15.7) | 756 (78.8) |
| PMS will reduce the prevalence of some genetic diseases | 112 (11.7) | 216 (22.5) | 632 (65.8) |
| Consanguinity can increase the risk of hereditary diseases | 6 (0.6) | 288 (30) | 666 (69.4) |
| PMS will reduce the prevalence of some sexual transmitted diseases | 72 (7.5) | 258 (26.9) | 630 (65.6) |
| PMS and genetic counselling should be confidential | 71 (7.4) | 331 (34.5) | 558 (58.1) |
| PMS cause psychological troubles to the couples | 371 (38.6) | 430 (44.8) | 159 (16.6) |
| The religious people should adopt the ideas of PMS in their discussion | 106 (11) | 238 (24.8) | 616 (64.2) |
| Law should obligate all future couples to do PMS and genetic counselling | 90 (9.4) | 115 (12) | 755 (78.6) |
| In detecting Sexually Transmitted Diseases, marriage decision must be left for freedom of the couple | 223 (24.3) | 345 (35.9) | 382 (39.8) |
| The medical counselling is important to be given after getting the results | 67 (7) | 276 (28.7) | 617 (64.3) |
| Any disease appeared in one of the couple has to be treated before marriage | 77 (8) | 209 (21.8) | 674 (70.2) |

As shown in Table 5, knowledge and perception scores have weak but statistically significant positive correlations with the students' stage. Attitude score shows no significant correlation with the students' stage. Regarding knowledge scores, significant differences exist between "less than 20" vs. "more than 22" and "20–22" vs. "more than 22" age groups, with the "more than 22" group scoring higher. Regarding perception scores, significant differences are found between "20–22" vs. "more than 22" age groups, with a near-significant difference between "less than 20" vs. "more than 22." Regarding attitude scores, significant differences exist between "20–22" vs. "more than 22" age groups. Other comparisons show no significant differences, as shown in Table 5.

The mean score of knowledge in females was (4.809) ± (1.696), (95% C.I: 4.671–4.947), while in males was (5.175) ± (1.739), (95% C.I: 4.999–5.351) with a P value of <0.001. Males have a higher mean knowledge score compared to females. The P value is less than 0.001, indicating a statistically significant difference in knowledge scores between genders.

Both genders have similar mean perception and attitude scores, and the P values were 0.383 and 0.273, respectively. This indicates no statistically significant difference in perception and attitude scores between genders.

Regarding marital status and knowledge score, single participants' mean score was (4.193) ± (1.714), (95% C.I: 4.801–5.026) while married individuals' mean score was (5.457) ± (1.759), (95% C.I: 5.038–5.876) and the (P value 0.007). This mean score of married individuals has a higher mean knowledge score than single individuals.

The mean perception score for single was (6.07) ± (4.5), (95% C.I: 5.78–6.37), and the mean for married was (7.47) ± (5.1), (95% C.I: 6.25–8.69) the (P value <0.001). This means that married individuals have a significantly higher mean perception score than single individuals.

The mean attitude score for singles was (1.402) ± (1.38), (95% C.I: 1.311–1.493), and the mean of married was (1.986) ± (1.749), (95% C.I: 1.569–2.403) and (P value <0.001) This

**Table 4. Attitude of the participants.**

| Question | Disagree (%) | Neutral (%) | Agree (%) |
|---|---|---|---|
| Carryout PMS | 50 (5.2) | 129 (13.4) | 781 (81.4) |
| Not- prefer relative marriage | 573 (59.7) | 318 (33.1) | 69 (7.2) |
| Appropriate time for PMS just before marriage | 40 (4.2) | 132 (13.8) | 788 (82.1) |
| Advise future couples to conduct PMS | 173 (18) | 202 (21) | 585 (60.9) |

**Table 5. The results of Kruskal Wallis test.**

| Kruskal Wallis test | Mean rank 1 | Mean rank 2 | Mean rank diff. | P value |
|---|---|---|---|---|
| **Knowledge Score and age group** | | | | |
| less than 20 vs. 20–22 | 457.8 | 471.1 | -13.35 | >0.999 |
| less than 20 vs. more than 22 | 457.8 | 562.0 | -104.2 | <0.001 |
| 20–22 vs. more than 22 | 471.1 | 562.0 | -90.85 | 0.001 |
| **Perception score and age group** | | | | |
| less than 20 vs. 20–22 | 478.7 | 460.1 | 18.55 | >0.999 |
| less than 20 vs. more than 22 | 478.7 | 538.0 | -59.35 | 0.067 |
| 20–22 vs. more than 22 | 460.1 | 538.0 | -77.90 | 0.008 |
| **Attitude score and age group** | | | | |
| less than 20 vs. 20–22 | 485.6 | 456.3 | 29.31 | 0.356 |
| less than 20 vs. more than 22 | 485.6 | 530.3 | -44.70 | 0.225 |
| 20–22 vs. more than 22 | 456.3 | 530.3 | -74.00 | 0.009 |
| Kruskal Wallis test | Mean rank 1 | Mean rank 2 | Mean rank diff. | P value |
| **Knowledge Score and age group** | | | | |
| less than 20 vs. 20–22 | 457.8 | 471.1 | -13.35 | >0.999 |
| less than 20 vs. more than 22 | 457.8 | 562.0 | -104.2 | <0.001 |
| 20–22 vs. more than 22 | 471.1 | 562.0 | -90.85 | 0.001 |
| **Perception score and age group** | | | | |
| less than 20 vs. 20–22 | 478.7 | 460.1 | 18.55 | >0.999 |
| less than 20 vs. more than 22 | 478.7 | 538.0 | -59.35 | 0.067 |
| 20–22 vs. more than 22 | 460.1 | 538.0 | -77.90 | 0.008 |
| **Attitude score and age group** | | | | |
| less than 20 vs. 20–22 | 485.6 | 456.3 | 29.31 | 0.356 |
| less than 20 vs. more than 22 | 485.6 | 530.3 | -44.70 | 0.225 |
| 20–22 vs. more than 22 | 456.3 | 530.3 | -74.00 | 0.009 |

shows that married individuals have a significantly higher mean attitude score than single individuals.

Both rural and urban residents have similar mean knowledge scores since the P value was 0.944, indicating no statistically significant difference in knowledge scores between resident areas. Both rural and urban residents have similar mean perception scores, and the P value is 0.892, indicating no statistically significant difference in perception scores between resident areas. Rural residents have a slightly higher mean attitude score than urban residents; the P value is 0.133, indicating no statistically significant difference in attitude scores between resident areas (Table 6).

## Discussion

The recent study reported that, the majority of the students had poor knowledge, only approximately quarter of them had good knowledge towards premarital screening test. This result is in line with regional findings. A recent population-based study conducted in Saudi Arabia revealed that while 52.4% of participants (n = 3283) had a fair understanding of PMS programs, only 9.2% of participants (n = 575) had satisfactory knowledge [1]. Similar results were obtained from another study conducted in Oman, which showed that although most participants (79%) had heard of PMS, half were ignorant of premarital testing [13]. Another study in Qatar reported low student knowledge of PMS [14]. Additionally, less than half, 139 (46.3%) respondents had a good understanding of PMS, as the study's results plainly showed [15]. This

**Table 6. Association knowledge, perception, and attitude with the demographics.**

| Variables | Knowledge score | | Perception score | | Attitude score | |
|---|---|---|---|---|---|---|
| | Mean ±(S.D) | 95% C.I | Mean ±(S.D) | 95% C.I | Mean ±(S.D) | 95% C.I |
| Female | 4.809 | 4.671–4.947 | 6.211 | 5.864–6.559 | 1.471 | 1.374–1.567 |
| Male | 5.175 | 4.999–5.351 | 6.119 | 5.617–6.622 | 1.405 | 1.231–1.579 |
| P value | <0.001 | | 0.383 | | 0.273 | |
| Single | 4.193 | 4.801–5.026 | 6.07 | 5.78–6.37 | 1.402 | 1.311–1.493 |
| Married | 5.457 | 5.038–5.876 | 7.47 | 6.25–8.69 | 1.986 | 1.569–2.403 |
| P value | 0.007 | | <0.001 | | <0.001 | |
| Rural | 4.868 | 4.358–5.378 | 6.83 | 6.038–7.622 | 1.774 | 1.446–2.101 |
| Urban | 4.958 | 4.846–5.07 | 6.137 | 5.835–6.439 | 1.426 | 1.333–1.519 |
| P value | 0.944 | | 0.892 | | 0.133 | |

result is consistent with a study from Ile-Ife, Nigeria, which found that 69% of respondents had an insufficient understanding of PMS [16]. 34.1% of respondents in a different survey on PMS knowledge and attitudes among students at the State School of Nursing in Sokoto, Nigeria, indicated that they were well-versed in the practice [17]. In addition, out of 492 participants, only 121 individuals (24.59%) possess sufficient genetic literacy. Additionally, according to the study, 262 respondents (60.16%) approved of the use of genetic screening [18].

These findings do highlight the necessity of health education initiatives to raise public awareness of PMS initiatives, which try to reduce the risk of certain genetic and sexually transmitted diseases among prospective spouses, particularly in nations where consanguineous marriages and inherited illnesses are highly common.

A recent study demonstrated that males have a higher mean knowledge score than females. However, the findings of some other studies indicated that women are more likely than men to have higher knowledge ratings. In Saudi Arabia and Kuwait, women were more knowledgeable than men [9,19]. A different Nigerian study [20] showed that female students' results differed from those of male pupils. Premarital genetic testing was more widely known among Syrian university students [21]. On the other hand, adult men in Oman knew more about premarital carrier screening than women, similar to the recent study [13]. However, investigations in Qatar did not find gender differences [22]. This could be because the target groups are different. The findings pertaining to gender differences in knowledge may differ, and this could be explained by socioeconomic and cultural factors. For example, because they are the focus of distinct education programs, women are traditionally better educated about PMS and marriage-related issues [23].

Relations between the fields in which students study can increase the knowledge score regarding PMS since the scientific or health and medical branches study much more about the different diseases; therefore, they know more about hereditary diseases than those who study other branches [23,24]. The existence of hereditary illnesses in families or personal illnesses, which help students cope with these circumstances and get more knowledge about them, is another topic that can broaden their knowledge [9,23]. This may be because families of affected individuals have access to tools like therapy sessions and educational materials that help them become more knowledgeable. Furthermore, people with a family history of a particular genetic issue are probably more curious to learn about the condition because they worry about how it can affect their families or future children.

According to a Kuwaiti study, partners who are not related to one another, such as cousins, know more about PMS. This could result from additional circumstances, such as parents' educational background, which probably increased their children's knowledge [9]. Furthermore,

compared to those who had heard of PMS, those who had never heard of it scored far worse on the knowledge test [14]. This is expected and necessitates student education initiatives and awareness efforts [23]. These initiatives, which can be carried out through academic courses and family members, which, in our survey, are rated as the source of PMS knowledge after academic courses and friends, are especially necessary for universities that are not health-related.

The information shows that the survey respondents answered various questions about the significance and consequences of PMS for prospective spouses. The participants' perspective on PMS. PMS is crucial for future couples, according to a great majority of 798 (83.1%), suggesting substantial support for its perceived importance. According to the survey, 41.4% of participants strongly agreed that the public would benefit more from genetic testing than suffer harm [14]. In a study conducted in Qatar, 28% of participants believed that PMS would be beneficial and avoid unanticipated consequences [22]. Given that 66.9% of participants in Bener et al.'s study had education levels below that of a university, the variation in the proportions is probably caused by the participants' educational backgrounds in the two studies. Additionally, a study evaluating young Jordanians' perceptions of premarital genetic testing revealed that about 90% of participants had a positive perception of premarital genetic screening because they thought it would help lower the likelihood of having affected children, which could be emotionally and financially taxing [2].

Moreover, a strong majority, 616 (64.2%), believe religious people should adopt PMS ideas in their discussions, showing support for integrating PMS into religious dialogue in the current study. Similarly, a study reported that it is interesting to note that 24.8% of participants, or nearly one-fourth, agreed that marriage should be prohibited if there is a possibility that the kid may be born with a genetic condition [14]. Additionally, a recent Omani study revealed that 36% of participants believed that laws and regulations should be put in place to stop high-risk marriages [25]. Though PMS is required in the Kurdistan region, couples have the option to stay married or dissolve their union at any time, provided they first attend a genetic counselling session wherein they are informed about the disease and its implications if their results indicate incompatibility and they sign a consent form attesting to their understanding of the session. The prohibition against marriage for high-risk couples presents a significant moral dilemma about the right to choose a mate. Moreover, there is no legal requirement to forbid the marriage of at-risk couples because there are alternative ways to reduce the likelihood of having an afflicted child, such as preimplantation genetic testing and prenatal diagnosis, which are becoming more widely available and accessible.

Of all participants, 776 show a poor attitude, accounting for 80.8% of the total. This group has the largest frequency by far, indicating that most participants are either disengaged or have unfavourable opinions regarding the topic at hand. 155 (16.1), a smaller but significant portion of individuals have a Fair Attitude. Merely 39% of the participants exhibit a Good Attitude, suggesting that a minuscule portion of the population harbours a positive perspective.

Given incompatible PMS findings, just one-third (37.4%) of the participants were willing to call off their marriage [14]. In additional research, this proportion was greater. For instance, in a study conducted at Taif University [26], in another study conducted in Saudi Arabia [23], it was 67.1%. In a population-based study conducted in Saudi Arabia [27], it was over 60%. As indicated by earlier studies [28], this might be because prenatal diagnosis and pregnancy termination (under certain conditions) are available in Qatar. In the current study, the attitude was not affected by any other factors. Other studies found that this may be a reflection of this highly educated population's level of religious knowledge and the significance of spreading unambiguous religious teachings endorsing PMS as a preventive measure for a healthy populace. According to a different survey conducted at Al-Taif University, only 5.2% of students disagreed that PMS taints fate [26].

This study was the first study conducted in Iraq and the Kurdistan region with a large sample size. It could be dependable for other studies and was key for educational and health programs. However, there are some limitations to the current study. Firstly, it could not be generalized to the whole population since only the university students participated in the study. Secondly, people answered the self-reported questionnaire in a way that demonstrated their commitment to the ideal scenario, even though it was subject to social decrement. This might have overstated the sentiment toward ending a marriage when the PMS results were mismatched. Third, self-selection bias might exist, and the sample may not represent the broader population fairly. Lastly, since the design is cross-sectional in nature, causal inferences should not be made.

## Conclusion

The participants widely acknowledge the importance and benefits of PMS; nevertheless, real knowledge regarding PMS is rather limited, and attitudes and perceptions are generally more negative than positive. The results emphasize the necessity of better education and awareness campaigns to increase understanding and modify attitudes and beliefs about PMS. In order to effectively address the important cultural component—particularly about the acceptance of marrying relatives—targeted educational initiatives may be necessary. Moreover, the notable distinctions in knowledge and attitude scores between married and single people imply that marital status and life experience are important factors in forming perspectives on PMS. Generally, for PMS to be more widely accepted and used, extensive education and the development of positive attitudes around it are crucial.

## Supporting information

**S1 Data. Dataset.**
(XLSX)

## Author Contributions

**Conceptualization:** Kochr A. Mahmood.

**Data curation:** Kochr A. Mahmood, Nazar P. Shabila.

**Formal analysis:** Kochr A. Mahmood, Nazar P. Shabila.

**Methodology:** Govand S. Sadraldeen, Abubakir M. Saleh.

**Project administration:** Govand S. Sadraldeen, Samir M. Othman.

**Writing – original draft:** Samir M. Othman, Nazar P. Shabila, Abubakir M. Saleh, Kameran H. Ismail.

**Writing – review & editing:** Kochr A. Mahmood, Nazar P. Shabila, Abubakir M. Saleh, Kameran H. Ismail.

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
