## [Decision Letter · Decision Letter 0]

2 Sep 2024

PGPH-D-24-01534

Knowledge, perception, and attitude toward premarital screening among university students in Kurdistan region- Iraq.

Dear Dr. Mahmood,

Thank you for submitting your manuscript to PLOS Global Public Health. After careful consideration, we feel that it has merit but does not fully meet PLOS Global Public Health’s publication criteria as it currently stands. Therefore, we invite you to submit a revised version of the manuscript that addresses the points raised during the review process.

The manuscript has been evaluated by two reviewers, and their comments are available below.

The reviewers have raised a number of concerns. They feel the manuscript would benefit from improving the introduction section with further detail and by providing the specific questions used in the questionnaire. The reviewers also recommend that you reorganise the results section to improve clarity and to enhance the impact of the discussion section by providing more in-depth discussion and explanation of your results.

Could you please carefully revise the manuscript to address all comments raised?

We look forward to receiving your revised manuscript.

Kind regards,

Johanna Pruller, Ph.D.

Staff Editor

PLOS 

Journal Requirements:

1. We would like to request copy editing.

Additional Editor Comments (if provided):

Reviewers' comments:

Reviewer's Responses to Questions

**Comments to the Author**

1. Does this manuscript meet PLOS Global Public Health’s publication criteria? Is the manuscript technically sound, and do the data support the conclusions? The manuscript must describe methodologically and ethically rigorous research with conclusions that are appropriately drawn based on the data presented.

Reviewer #1: Partly

Reviewer #2: Yes

2. Has the statistical analysis been performed appropriately and rigorously?

Reviewer #1: Yes

Reviewer #2: Yes

3. Have the authors made all data underlying the findings in their manuscript fully available (please refer to the Data Availability Statement at the start of the manuscript PDF file)?

Reviewer #1: Yes

Reviewer #2: Yes

4. Is the manuscript presented in an intelligible fashion and written in standard English?

Reviewer #1: Yes

Reviewer #2: Yes

5. Review Comments to the Author

Reviewer #1: Abstract :

Based on a preliminary review, the abstract appears well-structured and informative. It clearly outlines the study's objectives, methodology, key findings, and implications.

Introduction :

Clearly states the significance of PMS programs for public health

Emphasizes the need for university students to be informed about PMS

Mentions the limitations in current knowledge about student awareness of PMS

Links the study to the broader goal of improving future generations' well-being

Here are some suggestions for improvement:

• Briefly mention the specific research questions or hypotheses that the study aims to address.

• Consider providing a stronger justification for why university students were chosen as the target population for this study.

Method

This section provides a comprehensive overview of the methodological approach. However, some details could be further elaborated on in the full manuscript:

Regarding Questionnaire and Scoring System

1. Could authors please provide the specific questions used to assess participants' knowledge of premarital screening? It would be beneficial to include these questions in an appendix for reference.

2. Could authors elaborate on the development of the scoring system used to categorize participants' knowledge, perception, and attitude as poor, fair, or good? What specific criteria were used to determine these cut-off points?

Result

Combining tables in the Results section can improve readability and reduce redundancy. Explore using data visualization techniques like charts or graphs to represent complex data trends more effectively than tables.

Given that Table 3 provides a summary of key parameters, consider re incorporating the detailed data from Tables 4 and 5 into this on sequential display. This will improve the flow of the results section and make it easier for readers to understand the key findings.

Table 6 and Table 7 can be simplified. What is the basis for determining the age of 20 and 22? I think it is not very relevant for the age limit to be displayed as one of the parameters. Class level is more meaningful.

While it's important to summarize key findings from tables in the text, avoid simply restating the data verbatim. Focus on interpreting the results and explaining their significance. Consider using descriptive statistics (means, standard deviations, percentages) and effect sizes to convey the data concisely and effectively. This will help to improve the readability of the results section and strengthen the overall impact of your findings.

The Results section should focus on presenting the most critical findings that directly address the research questions. Avoid overwhelming readers with excessive statistical detail. Instead, prioritize key results and use tables or figures to supplement the text.

Discussion

The Discussion section provides a good overview of the study's findings in relation to existing literature. To enhance its impact, consider the following suggestions:

Strengthen the Argument and Interpretation

• Deepen the Interpretation: Explore the underlying reasons for the observed differences in knowledge, perception, and attitude between groups (e.g., gender, marital status, residential area). Consider potential sociocultural, economic, or educational factors influencing these disparities.

• Theoretical Framework: If applicable, integrate a theoretical framework to provide a more structured interpretation of the findings. For instance, a theory of planned behavior or health belief model could help explain the relationships between knowledge, perception, and attitude.

• Comparative Analysis: Expand the comparison with previous studies to include more recent and relevant research. Highlight the unique contributions of your study to the existing body of knowledge.

do not repeat result section into discussion (like in the first paragraph).

Why do your findings align with those of other studies conducted in the same region?

• What are the potential underlying factors contributing to this consistency?

• How can you provide a more in-depth analysis to explain these similarities beyond simply stating the concurrence?

You mention : “The findings pertaining to gender differences in knowledge may differ, and this could be explained by socioeconomic and cultural factors. For example, because they are the focus of distinct education programs, women are traditionally better educated about PMS and marriage-related issues”. Please add more reasonable finding to strengten the remark. You can compare with another region on similar finding and please adjust the potential reasoning, add https://doi.org/10.37268/mjphm/vol.20/no.3/art.407

While the studies cited demonstrate a generally positive perception of PMS, further research is needed to explore the nuances of these attitudes across different cultural and socioeconomic contexts.

The high level of support for PMS among participants underscores the importance of investing in public education campaigns to increase awareness and uptake of these services.

It would be beneficial to compare the findings of this study with those from other regions to identify global trends in attitudes towards premarital screening.

Overall comment for discussion section: To enhance the depth and impact of your findings, consider providing more in-depth explanations and interpretations for each statement on each paragraphs. This will help to reveal the underlying significance of the results and contribute to a more comprehensive understanding of the research topic.

Conclusion

The conclusion should clearly state the main findings, emphasize the need for improved education on PMS, and suggest potential areas for future research.

Reviewer #2: The manuscript meets the basic requirements for publication in PLOS Global Public Health, with technically sound research and data that supports the conclusions. However, I recommend minor revisions to address the following points:

- there are minor issues such as the lack of discussion on the reliability and validity of the questionnaire and there is a minor inconsistency in the reported sample size. Initially, it mentions a sample size of 500, but the study ended with 960 respondents. Clarifying this discrepancy and explaining the larger-than-expected response rate would be beneficial.

- Improve the manuscript’s clarity and readability through grammatical corrections, simplified sentence structures, and consistent terminology.

6. PLOS authors have the option to publish the peer review history of their article (what does this mean?). If published, this will include your full peer review and any attached files.

**Do you want your identity to be public for this peer review?** For information about this choice, including consent withdrawal, please see our Privacy Policy.

Reviewer #1: No

Reviewer #2: **Yes: **SUHA SULIMANI

---

## [Decision Letter · Decision Letter 1]

28 Oct 2024

Knowledge, perception, and attitude toward premarital screening among university students in Kurdistan region- Iraq.

PGPH-D-24-01534R1

Dear Dr Mahmood,

We are pleased to inform you that your manuscript 'Knowledge, perception, and attitude toward premarital screening among university students in Kurdistan region- Iraq.' has been provisionally accepted for publication in PLOS Global Public Health.

Best regards,

Julia Robinson

Executive Editor

Reviewer Comments (if any, and for reference):

Reviewer's Responses to Questions

**Comments to the Author**

1. If the authors have adequately addressed your comments raised in a previous round of review and you feel that this manuscript is now acceptable for publication, you may indicate that here to bypass the “Comments to the Author” section, enter your conflict of interest statement in the “Confidential to Editor” section, and submit your "Accept" recommendation.

Reviewer #1: All comments have been addressed

Reviewer #2: All comments have been addressed

2. Does this manuscript meet PLOS Global Public Health’s publication criteria? Is the manuscript technically sound, and do the data support the conclusions? The manuscript must describe methodologically and ethically rigorous research with conclusions that are appropriately drawn based on the data presented.

Reviewer #1: Yes

Reviewer #2: Yes

3. Has the statistical analysis been performed appropriately and rigorously?

Reviewer #1: Yes

Reviewer #2: Yes

4. Have the authors made all data underlying the findings in their manuscript fully available (please refer to the Data Availability Statement at the start of the manuscript PDF file)?

Reviewer #1: Yes

Reviewer #2: Yes

5. Is the manuscript presented in an intelligible fashion and written in standard English?

Reviewer #1: Yes

Reviewer #2: Yes

6. Review Comments to the Author

Reviewer #1: Thanks, the article have meet standard criteria for publication. Good luck with other publications.

Reviewer #2: (No Response)

7. PLOS authors have the option to publish the peer review history of their article (what does this mean?). If published, this will include your full peer review and any attached files.

**Do you want your identity to be public for this peer review?** For information about this choice, including consent withdrawal, please see our Privacy Policy.

Reviewer #1: No

Reviewer #2: No
